# Microstructure and Tribo-Behavior of WC–Cr_3_C_2_–Ni Coatings by Laser Cladding and HVAF Sprayed: A Comparative Assessment

**DOI:** 10.3390/ma16062269

**Published:** 2023-03-11

**Authors:** Ziying Zhang, Weizhou Li, Ruixia Yang, Xiaolian Zhao, Houan Zhang

**Affiliations:** 1School of Resources, Environment and Materials, Guangxi University, Nanning 530004, China; 2School of Materials Science and Engineering, Xiamen University of Technology, Xiamen 361024, China

**Keywords:** WC–Cr3C2–Ni, HVAF-WC coating, LC-WC coating, wear resistance

## Abstract

SK5 steel is the base material used for the preparation of the wrinkle scraper, whose service life strongly affects the working efficiency and economic benefits. In this work, WC–Cr3C2–Ni coating was deposited on the SK5 steel substrate by using High-velocity air fuel spray (HVAF) and Laser cladding (LC) processes respectively, named HVAF-WC coating and LC-WC coating. The microstructure and wear resistance of both coatings were analyzed, and were compared with the substrate sample. Results showed that the coatings were adhesive well onto the substrate. More WC with fine crystals is retained in HVAF-WC coating due to low flame flow temperature, while WC of LC-WC coating is characterized by columnar crystals. The wear rate of HVAF-WC and LC-WC coating was 4.00 × 10^−7^ mm^3^/(N•m) and 3.47 × 10^−6^ mm^3^/(N•m), respectively, which was two and one orders of magnitude lower than SK5 steel with 3.54 × 10^−5^ mm^3^/Nm. HVAF-WC coating exhibited the best wear resistance because of significant fine grain strengthening, which wear mechanism is mainly dominated by abrasive wear. Thus, it was thought that HVAF-WC coating is more effective ways to improve the wear resistance of SK5 steel, comparing with LC-WC coating.

## 1. Introduction

High-speed toilet paper machines are the driving force behind the production of household paper. As a critical component of high-speed toilet paper machine, wrinkle scraper comes in direct contact with the paper and bears severe friction, shock and vibration, whose service plays a great role in production efficiency. SK5 steel with a high hardness about 57–62 HRC is the most commonly used base material for the wrinkle scraper. However, wear resistance is still insufficient for the repeated high-speed friction, resulting in a frequent replacement of the scrapers, so as to reduce the production efficiency. Therefore, seeking a suitable surface coating to improve wear resistance of SK5 steel has become an urgent need to be served. Tungsten carbide (WC) is an important class of wear-resistance material and can be well adjusted by metal melts such as Co, Fe, and Ni, which are widely used to prepare wear-resistance coatings on the surface of mechanical parts [1,2,3,4].

High-velocity air fuel spray (HVAF) is a coating deposition technology developed in recent years, which fabricates coating by high velocity flying particles flux through Laval nozzle under compressed air and fuel burning in the combustion chamber, and nitrogen carrying powder material into flame flow [5,6]. Compared to traditional High-velocity oxygen fuel spray (HVOF), HVAF produces lower flame temperatures and higher particle velocities [7,8,9]. Differently, laser cladding (LC) produces coating rapidly under high energy density laser beam and solidifying on the protected substrate with a metallurgical bonding [10,11].

By HVAF or LC technology, WC coatings with different metallic elements were deposited on steel substrate with a marked improvement in tribological properties. For example, Bhosale et al. [2] deposited WC–Cr3C2–Ni coatings on SS 316L substrate by plasma spray (APS) and HVOF, and found that the average coefficient of friction (COF) for APS coating, HVOF coating, and SS 316L respectively varies from 0.171 to 0.234, 0.151 to 0.178, and 0.456 to 0.536. Both coatings have significant improvement in the wear resistance over SS 316L, and HVOF coating exhibited lower wear rates than APS coating due to relatively better retained hard WC phase. Bolelli et al. [12] sprayed WC–10Co4Cr coatings onto carbon steel substrates using HVOF and HVAF and found that all samples obtained mild wear rates of <10^−7^ mm^3^/(Nm) and COF of ≈0.5. Moreover, HVOF cause higher carbon loss, and thus lower WC content in the coatings. Hu et al. [3] deposited WC composite coating on surface of 5Cr5MoSiV1 TBM cutter ring by laser cladding and found that wear resistance of the coating was about 7 times higher than that of the substrate, attributed to hard phase WC widely distributed in the coating effectively hindering the pressing and ploughing of hard rock particles. Wei et al. [13] produced NiCrAl-WC coatings on AISI H13 steel by laser cladding and found that and found that the coating wear rate was decreased with the increasing WC content. The NiCrAl-40%WC coating had the best wear resistance, which average COF was 0.379 and wear rates was 28.0 μm^3^ N^–1^ mm^–1^, respectively decreased by 17.8% and 67.5% compared with the substrate.

From the above research, it was known that WC–Cr3C2–Ni is one of the most important commercially powder to prepare high-wear resistant coatings for steel substrate [14,15,16]. Herein Cr_3_C_2_ was strengthening phase and metal Ni worked as binding phase. However, the effect of the preparation method on the structure and properties of WC-based coatings has not been compared in previous studies. Thus, we chose this powder as the raw material to prepare the coating on the surface of SK5 steel by using the HVAF and LC methods, and compared the tribological behavior of two coatings with the substrate.

## 2. Experimental

### 2.1. Coating Materials and Processes

WC–Cr3C2–Ni powder (Chongyi ZhangYuan Tungsten Co., Ltd., Ganzhou, China. 5–35 μm) was used as raw material for coating deposition and SK5 steel (Dongguan Fenmei Metal Materials Co., Ltd., Dongguan, China) was substrate material, which chemical composition are shown in Table 1. The substrates were cut into cuboids with the size of 100 × 100 × 8 mm^3^ by a wire electrode cutting, and then were ground with graded silicon carbide sandpapers. After that, the samples were put into ethanol for ultrasonic cleaning to remove dirt, and dried in air for standby.

HVAF-WC coating was deposited by commercial HVAF thermal spraying M3 system (Uniquecoat Technologies LLC, Oilville, VA, USA). Spraying parameters were listed in Table 2. It should be noted that, in this study, the parameters used in both coatings were optimized after preliminary pre-study, with the purpose of better research and comparison. Before deposition, all specimens were mounted on a rotating sample holder, and then were grit-blasted by using corundum particles. Subsequently, the specimens were preheated to reduce the oxidation effects on a surface during spraying.

LC-WC coating was prepared by the ZKSX-2004 laser system, which was equipped with a 2.0 kW solid-state laser with fiber-optic transmission and paraxonic powder feeding system. The wave length of the laser beam was 1064 nm. Before laser cladding processing, the specimens were preheated to 300 °C to reduce the residual stress. Optimized laser cladding parameters were listed in Table 3 based on preliminary experiments.

### 2.2. Microstructural Characterization

The phases of powder feedstock, HVAF- and LC-WC coating samples were identified by using a Siemens X-ray diffractometer (XRD, MiniFlex600-C, Rigaku Corporation, Tokyo, Japan) with Cu Ka radiation performed at 40 kV and 40 mA (1.5406 Å, step size of 0.01°, 2θ scanning range of 20–85 and scanning speed of 10°/min). Cross-section of the coated specimens were ground with graded emery papers and polished with 1.0 μm diamond paste to produce a surface roughness (Ra) of 0.10 μm. A scanning electron microscopy (SEM, TM4000, Hitachi, Tokyo, Japan) equipped with energy dispersive spectrometry (EDS, EDAX Genesis, Ametek, Berwyn, PA, USA) was used to analyze the morphologies and chemical compositions of the powder and coatings. The amounts of voids in the coatings was calculated by Image Pro Plus of SEM × 1000 images of the coating cross sections. For accuracy in measurement mean value ten micrographs were taken. The chemical bonding states of coatings were investigated using X-ray photoelectron spectroscopy (XPS) (ESCALAB 250XI, Thermo, Waltham, MA, USA) with monochromatic Al Kα radiation (1486.6 eV) at 400 W and 14 kV.

### 2.3. Microhardness and Wear Testing

A Vickers hardness tester (HVS-1000Z, Shanghai Zhongyan Instrument Manufacturing Factory, Shanghai, China) was used to test the surface microhardness of coatings and substrate under 300 gf load and dwell time of 15 s. For accuracy in measurement mean value ten readings was taken. Before the wear test, the surface of the substrate, HVAF- and LC-WC coating samples was polished to produce a surface roughness (Ra) of 0.10 μm and then cleaned with alcohol. Ball-on-disk friction and wear test was conducted on a HT-1000 high-temperature friction and wear test machine (Lanzhou Zhongke Kaihua Science and Technology Development Co., Ltd., Lanzhou, China) in air at room temperature, where the load, sliding speed, sliding time and diameter of sliding track were fixed at 560 rpm, 60 min and 5.0 mm, respectively, and silicon-nitride (Si_3_N_4_) ceramic balls with a diameter of 4 mm and hardness of 90 HRC were used as the friction pair. After the wear test, the profiles, morphologies and chemical elements of wear tracks were analyzed using three-dimensional non-contact surface mapping profile surface (Dektak XT, Bruker, Billerica, MA, USA), SEM and EDS. The wear rate (*W*) were calculated using following equation [11,17]:(1)$$W=\frac{2\pi RA}{FS}$$
where “*R*” is the radius of the wear track, “*A*” is the cross-section area of the wear track, “*F*” is the normal load, and “*S*” is the total sliding distance.

## 3. Results

### 3.1. Phase Compositions and Microstructures

Figure 1 reveals the SEM morphology, EDS analysis diagram and particle distribution of WC–Cr3C2–Ni powder. As shown in Figure 1a, the powder exhibits spherical shapes of various sizes and no agglomeration was found between the particles. Figure 1b shows the particle size distribution of WC–Cr3C2–Ni feedstock powder: Dv(10) = 13.6 μm, Dv(50) = 19.9 μm, and Dv(90) = 28.2 μm. Figure 1c depicts EDS spectra of precursor powder, with W element as the main component and a certain amount of elements Cr, Ni and C.

Figure 2 presents XRD patterns of WC–Cr3C2–Ni powder, HVAF-WC coating, and LC-WC coatings. From XRD pattern of the powder, it is found mainly composed of WC, Cr_3_C_2_ and Ni phase in the raw powder. The XRD pattern of HVAF-WC coating shows that the coating is also composed by hard phase WC and Cr_3_C_2_, and metal phase Ni, which indicates that no phase transformation occurs during the spraying process. Hard phase is well preserved in favor of wear resistance. For the LC-WC coating, it is composed of WC, Cr_2_O_3_ and Ni, and no peaks of Cr_3_C_2_ phase are detected. It was because oxidation reaction occurring during the cladding process. The absence of hard phase Cr_3_C_2_ is detrimental to wear resistance. Moreover, WC peaks of the HVAF-WC coating are significantly stronger than that of the LC-WC coating, indicating that more WC hard phase is retained in the HVAF-WC coating.

Figure 3 reveals cross-section morphology of HVAF-WC coating. As shown in Figure 3a, SEM × 300 images, average thickness of the coating is 180 μm, which was adhered well onto the substrate. No obvious cracks and lamellar structure are observed, as shown in the arrows in Figure 3b, SEM × 2500 images, which is essential for improving the tribological properties. Only a few pores are in the coating, and the porosity is 0.50%. The coatings contain white phases and dark gray phases. EDS analyses (Table 4) of yellow point in Figure 3b reveal that white phases are rich in W element and gray phases are rich in Cr element. Combined with the XRD, it is speculated that the white phases are mainly WC, and the gray phases are mainly Cr_3_C_2_. WC phase presents dense fine crystals, related to the ability of Cr_3_C_2_ to inhibit grain growth [14,18].

Figure 4 shows cross-section morphology of the LC-WC coating. As shown in Figure 4a, SEM ×50 images, the coating is also adhesive well onto the substrate, with a thickness of 1250 μm. No cracks but round pores (black arrows in Figure 4a) are clearly visible in the coating, and porosity is 2.30%. From the Figure 4b, SEM × 2500 images, WC phase presents columnar crystals, which size is larger than that of HVAF-WC coating. Table 5 shows EDS analysis of the marked positions in Figure 4. It is worth noting that Fe element is detected in grey regions, indicating that the Fe in the substrate was diluted into the LC-WC coating and acts as a bonding phase [19].

### 3.2. XPS Analysis

The XPS analyses of the HVAF-WC coating sample (Figure 5) and LC-WC coating sample (Figure 6) were further carried out to confirm the chemical composition. As shown in Figure 5a, the peaks of W 4f, Cr 2p, Ni 2p, C 1s, and O 1s were detected. The refined spectrum manifested that the W4f spectrum with four peaks located at a binding energy of 31.80 eV, 34.10 eV, 35.40 eV and 37.40 eV, which are arising from W 4f_7/2_ orbitals of WC, W 4f_5/2_ orbitals of WC, W 4f_7/2_ orbitals of WO_3_, and W 4f_5/2_ orbitals of WO_3_, respectively (Figure 5b). As shown in Figure 5c, four fitting peaks at 574.20 eV, 576.80 eV, 583.60 eV and 586.40 eV for Cr 2p spectrum, which are arising from Cr 2p_3/2_ orbitals of Cr_3_C_2_, Cr 2p_3/2_ orbitals of Cr_2_O_3_, Cr 2p_1/2_ orbitals of Cr_3_C_2_, and Cr 2p_1/2_ orbitals of Cr_2_O_3_, respectively [20]. In refined spectrum of Ni 2p (Figure 5d), six fitting peaks at 852.73 eV, 856.00 eV, 861.00 eV, 870.00 eV, 874.40 eV and 879.10 eV, which are arising from Ni 2p_3/2_ orbitals of Ni, Ni 2p_3/2_ orbitals of NiO, Ni 2p_3/2, sat_ orbitals of NiO, Ni 2p_1/2_ orbitals of Ni, Ni 2p_1/2, sat_ orbitals of Ni, and Ni 2p_1/2, sat_ orbitals of NiO respectively.

For LC-WC coating, the peaks of W 4f, Cr 2p, Ni 2p, C 1s, and O 1s were detected (Figure 6). As shown in Figure 6b, the refined spectrum manifested W4f spectrum with four peaks at 31.50 eV, 33.70 eV, 35.20 eV and 37.40 eV, which are arising from W 4f_7/2_ orbitals of WC, W 4f_7/2_ orbitals of W, W 4f_7/2_ orbitals of WO_3_, and W 4f_5/2_ orbitals of WO_3_, respectively. The refined spectrum manifested that the Cr 2p spectrum with two peaks located at a binding energy of 576.80 eV, 577.30 eV, and 586.40 eV, which corresponded to the Cr 2p_3/2_, Cr 2p_3/2_, and Cr 2p_1/2_ orbitals for Cr_2_O_3_ respectively [20] (Figure 6c). No obvious peaks are detected in refined spectrum of Ni 2p (Figure 6d).

### 3.3. Wear and Friction Behavior of SK5 Steel, LC-WC Coating and HVAF-WC Coating Samples

Figure 7 shows the COF-distance curves of HVAF-WC coating, LC-WC coating, and SK5 steel samples. Note that the COFs of all samples increase sharply at the initial stage, it is called “run-in stage”, lasted about 5 min, where all surfaces are cleaned from contaminations and polished [17]. After the run-in stage, the COFs of both coatings gradually tend to be relatively stable, “steady-state stage” begins. But for SK5 steel, the COF is still fluctuating greatly during “steady-state stage”. This indicates that coating effectively reduces the degree of wear of the SK5 steel. The corresponding average COFs during steady-state stage for HVAF-WC coating, LC-WC coating and SK5 steel are calculated, which is 0.50, 0.66 and 0.63, respectively. Moreover, the average friction force (N) of HVAF-WC coating, LC-WC coating and SK5 steel in the friction and wear process is 7.44, 9.87 and 9.39, respectively. 

Figure 8 shows surface microhardness, wear rate and two-dimensional (2D) morphology of wear test samples. Before the wear test, the surface microhardness of HVAF-WC coating, LC-WC coating, and SK5 steel samples were tested, the results as shown in Figure 8a, which is 1208.57 ± 49.30 HV0.3, 1018.25 ± 36.05 HV0.3, and 130.22 ± 6.66 HV0.3, respectively. The SK5 steel exhibits the highest wear rate of 3.54 × 10^−5^ mm^3^/Nm, the wear rate of LC-WC coating and HVAF-WC coating is 3.47 × 10^−6^ mm^3^/Nm and 4.00 × 10^−7^ mm^3^/Nm, which is one and two orders of magnitude lower than SK5 steel, respectively. Compared with WC–Cr3C2–Ni coating deposited by HVOF [2], HVAF-WC coating in this study displays displayed lower wear rates. Negative correlation between wear rate and microhardness can be found, the higher microhardness, the lower wear rate. Figure 8b–d reveals the typical 2D morphology from the corresponding wear track profiles. With minimum depth of HVAF-WC coating as 0.5 μm, intermediate of LC-WC coating as 7.1 μm and maximum of SK5 steel as 30.7 μm.

Further details about the worn surface morphologies of HVAF-WC coating (Figure 9a,b), LC-WC coating (Figure 9c,d) and SK5 steel sample (Figure 9e,f) are exhibited in Figure 9.

As shown in Figure 9a, the wear track of HVAF-WC coating is not obviously. There are no cracks and large holes in the surface morphology. Figure 9b shows a detailed worn surface micrograph. No obvious scratches are observed in the worn area. Due to a large number of WC characterized fine grain evenly distributed in the metal matrix, Si_3_N_4_ counterpart was difficult to be pressed into the HVAF-WC coating surface for tangential movement, thereby greatly improving the wear resistance. WC particles and metal matrix provide strong cohesive force due to few defects of HVAF-WC coating. As a result, WC is difficult to peel off even under repeated friction, only being constantly scratched. The EDS analysis of wear track (red box A position in Figure 9b shows 1.43C-25.28O-14.31Cr-2.29Ni-56.69W (wt.%). Accordingly, wear mechanism of the HVAF-WC coating is mainly dominated by abrasive wear caused by hard phases, accompanied with oxidative wear.

For the LC-WC coating sample, cracks (orange arrows in Figure 9c) and holes (green arrows in Figure 9c) are clearly visible on worn surface micrograph. As shown in Figure 9d, worn surface of LC-WC coating was characterized by furrows, local plastic deformation of the metallic matrix, and fragmentation and removal of WC microparticles. Counter body Si_3_N_4_ ball is cut into LC-WC coating by applied load, which left the furrows on the worn track. Defects (cracks and pores) results in micro-cutting of the metallic matrix and fracture of WC during sliding friction and wear test. The EDS analysis of the red box B position in Figure 9d results in 1.32C-15.79O-16.85Cr-0.62Ni-65.42W (wt.%). As such, plastic deformation of metal matrix and peeling of WC hard particles are the main material removal mechanisms, and abrasive wear and oxidative wear are the main wear mechanisms for LC-WC coating.

For SK5 steel, the wider wear track is generated (Figure 9e). Due to the low hardness (130 ± 3.0 HV0.3), SK5 steel cannot prevent the intrusion and ploughing of Si_3_N_4_ ball, and has undergone a series of serious plastic deformation. A large number of grooves (orange arrows in Figure 9f) parallel to the sliding direction are distributed on the worn surface, which attributed to the sliding of hard particle debris trapped between the SK5 steel surface and the Si_3_N_4_ counter body. In addition, delamination phenomenon (blue arrows in Figure 9f) and oxide wear debris (green arrows in Figure 9f) are observed in the area of the wear track. EDS analysis (red box C in Figure 9f) shows 1.40C-15.99O-1.64Si-80.97Fe (wt.%). Counterpart element Si was transferred to the wear track. Therefore, adhesive wear and oxidative wear are the dominant wear mechanisms of the SK5 steel.

## 4. Discussion

### 4.1. Formation of HVAF-WC Coating and LC-WC Coating

A schematic model of coatings formation can be suggested based on the microstructure observation, as shown in Figure 10. At the beginning, the WC–Cr3C2–Ni powder was placed into the powder delivery system. This powder is atomized by nitrogen gas and characterized by different particles–it contains hard phase WC, hard phase Cr_3_C_2_, and bonding phase metal Ni. Throughout the spraying process metal Ni melt completely and Cr_3_C_2_ undergo partial melting, while WC does not melt. Thus, the melt is enriched with Ni, Cr and C. The solidification begins around the solid WC particles. Eventually, the WC particles exist in the coating in a fine-crystalline state. During the laser processing Ni and Cr_3_C_2_ melt completely while WC undergo partial melting only. Therefore, the melt is rich in Ni, Cr, and a few W elements. Partially melted WC particles gather together to solidify, forming a structure of columnar crystals.

### 4.2. Analysis of Friction and Wear Process

To gain a more visual understanding of the wear mechanism, schematic models of HVAF-WC coating, LC-WC coating, and SK5 steel samples, as shown in Figure 11. Figure 11a shows schematic diagrams of wear mechanism for HVAF-WC coating. WC particles characterized by small size facilitates fine grain strengthening. And the spacing among the WC particles is small, enhancing the dispersion strengthening. In this case, even when the binder is removed, the exposed hard particles are hardly pulled out because of the dragging effect from the cluster embedded in the binder below [21]. Meanwhile, the high hardness of the coating effectively resists the pressure of the grinding ball, the grooves are narrow and shallow, and the scratches on the wear track are smoother. Therefore, the wear resistance of the HVAF-WC coating can be significantly improved. Figure 11b shows schematic diagrams of wear mechanism for LC-WC coating. Large areas of the binder have nearly been removed, so that the WC particles are exposed to the surface. In this case, the isolated WC particles are very likely to be separated. Unlike the fine grain structure of HVAF-WC coating, WC of LC-WC coating is characterized by column crystal with larger size, which are easily to be broken under load and be peeled off. The LC-WC coating material is separated layer by layer.

## 5. Conclusions

WC–Cr_3_C_2_–Ni coatings were prepared on SK5 steel by high-velocity air fuel (HVAF) and laser cladding (LC), named LC-WC coating and HVAF-WC coating, which microstructure and tribological properties comparatively were investigated. The results summarized as follows:Both coatings are adhesive well with substrate, no cracks at the coating/substrate interface. More WC phases with fine crystals are retained in HVAF-WC coating due to low flame flow temperature, while WC of LC-WC coating is characterized by columnar crystals. Porosity of HVAF-WC and LC-WC coating is 0.50% and 2.30%, respectively.Microhardness of HVAF-WC coating is 1208.57 ± 49.3 HV0.3, higher than LC-WC coating with 1018.25 ± 36.05 HV0.3, which is associated with lower porosity, more WC phases, fine grain strengthening and dispersion strengthening.Average COF of HVAF-WC coating, LC-WC coating and SK5 steel is 0.50, 0.66 and 0.63, respectively. Negative correlation between wear rate and microhardness was found. The wear rate of HVAF-WC and LC-WC coating is 4.00 × 10^−7^ mm^3^/(N•m) and 3.47 × 10^−6^ mm^3^/(N•m), respectively, which is two and one orders of magnitude lower than SK5 steel with 3.54 × 10^−5^ mm^3^/Nm, respectively. The dominant wear mechanism of both coating is abrasive wear, while the SK5 steel is adhesive wear and oxidative wear.

Therefore, the present work demonstrates that WC–Cr3C2–Ni coatings are useful for improving the wear resistance of the SK5 steel. Compared with LC-WC coating, the HVAF-WC coating had fewer defects, a higher microhardness, and a better wear resistance.

## Figures and Tables

**Figure 1 materials-16-02269-f001:**
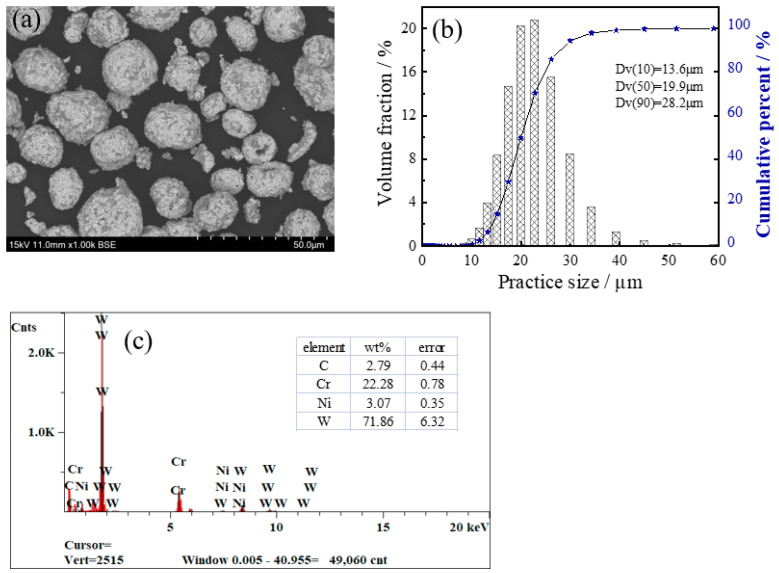
SEM morphology (**a**), particle distribution (**b**), and EDS diagram (**c**) of WC–Cr3C2–Ni powder.

**Figure 2 materials-16-02269-f002:**
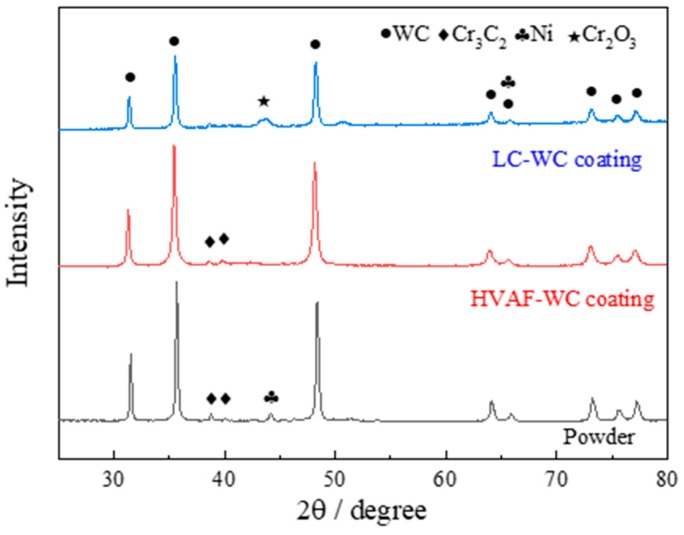
XRD patterns of WC–Cr_3_C_2_–Ni powder, HVAF-WC coating and LC-WC coating.

**Figure 3 materials-16-02269-f003:**
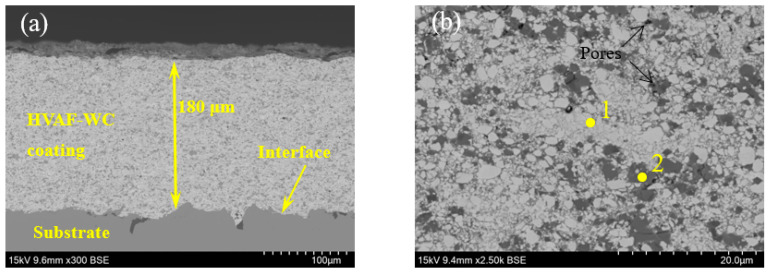
Cross-section morphology of HVAF-WC coating: (**a**) lower magnification, (**b**) higher magnification.

**Figure 4 materials-16-02269-f004:**
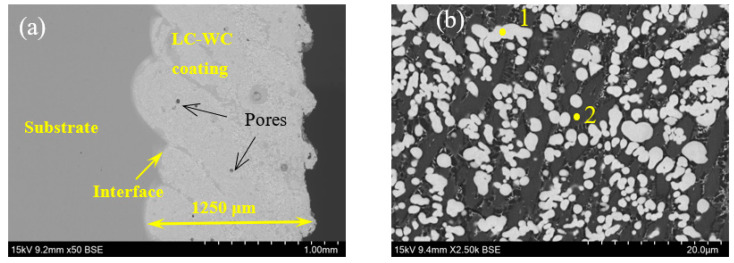
Cross-section morphology of LC-WC coating: (**a**) lower magnification, (**b**) higher magnification.

**Figure 5 materials-16-02269-f005:**
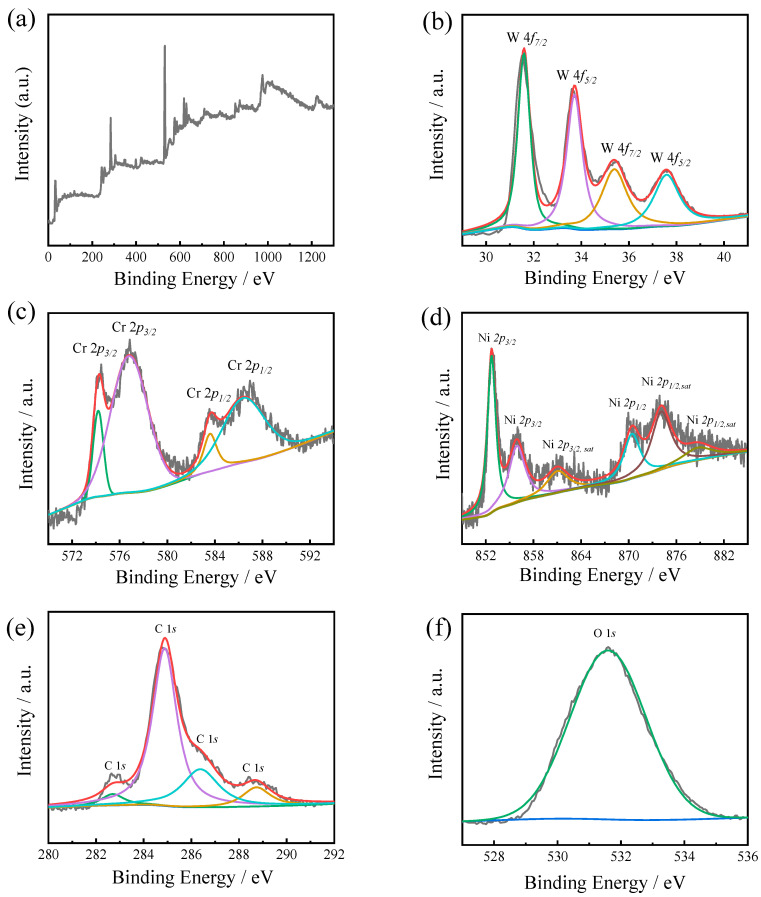
XPS survey spectra (**a**) and refined spectra of W 4f (**b**), Cr 2p (**c**), Ni 2p (**d**), C 1s (**e**), and O 1s (**f**) for HVAF-WC coating.

**Figure 6 materials-16-02269-f006:**
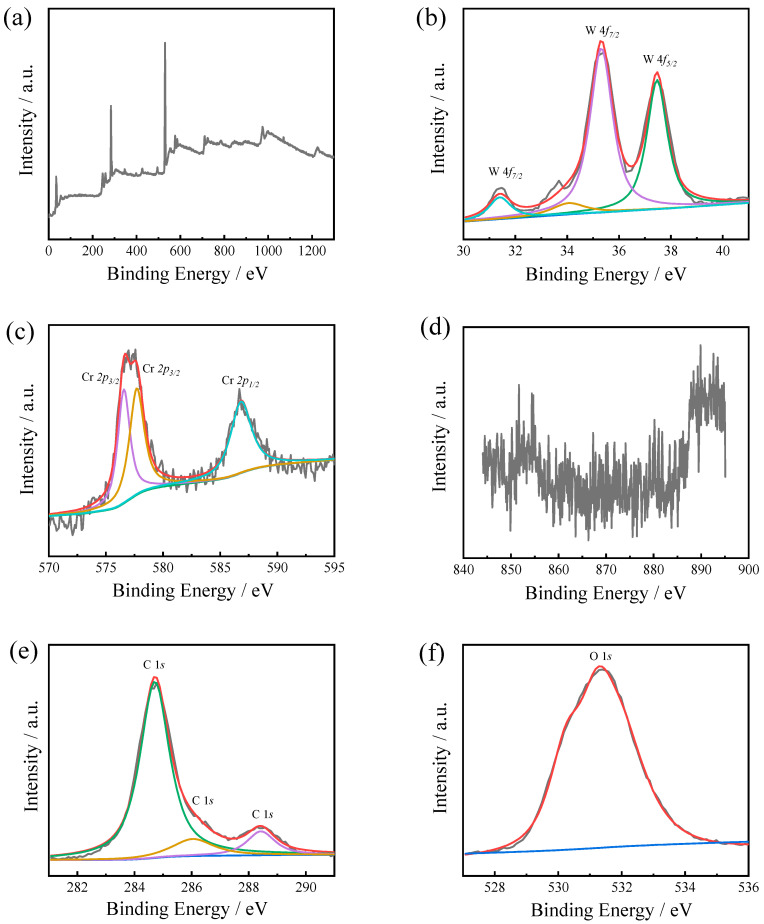
XPS survey spectra (**a**) and refined spectra of W 4f (**b**), Cr 2p (**c**), Ni 2p (**d**), C 1s (**e**), and O 1s (**f**) for LC-WC coating.

**Figure 7 materials-16-02269-f007:**
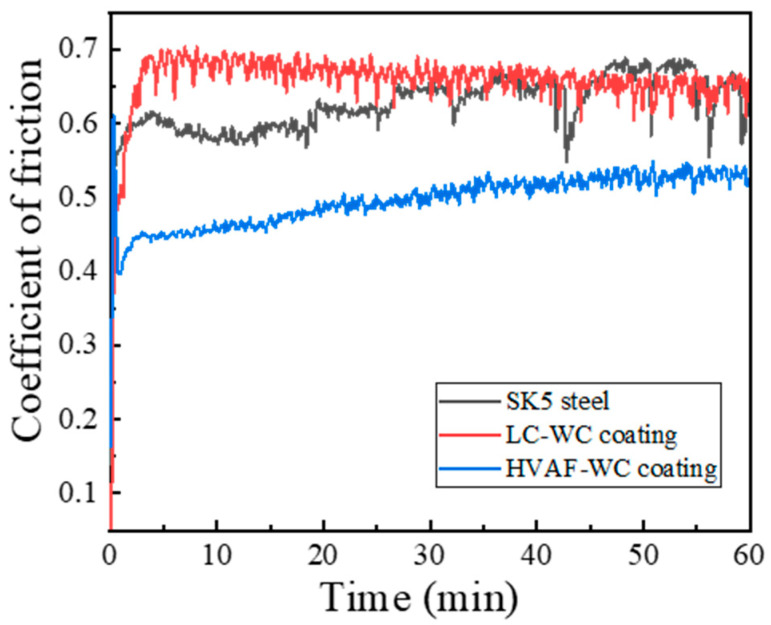
COF-distance curves of HVAF-WC coating, LC-WC coating and SK5 steel samples.

**Figure 8 materials-16-02269-f008:**
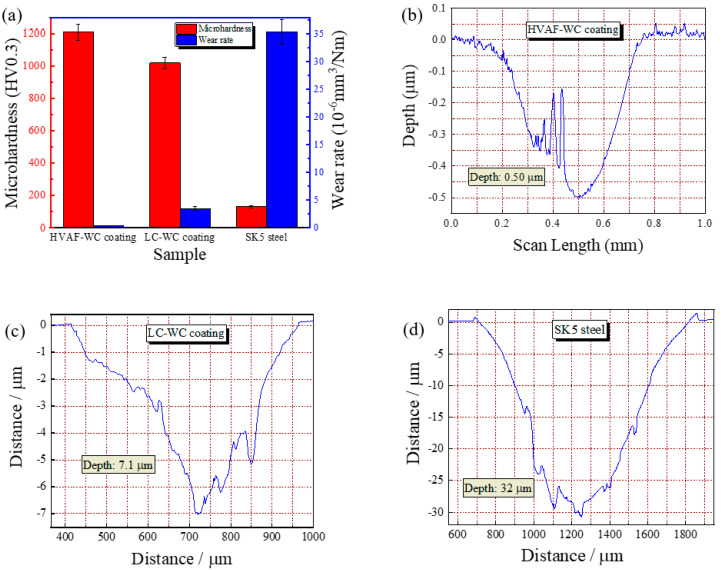
Surface microhardness, wear rates and 2D morphology of wear test samples, (**a**) Surface microhardness and wear rates; (**b**) 2D morphology of HVAF−WC coating; (**c**) 2D morphology of LC−WC coating; (**d**) 2D morphology SK5 steel.

**Figure 9 materials-16-02269-f009:**
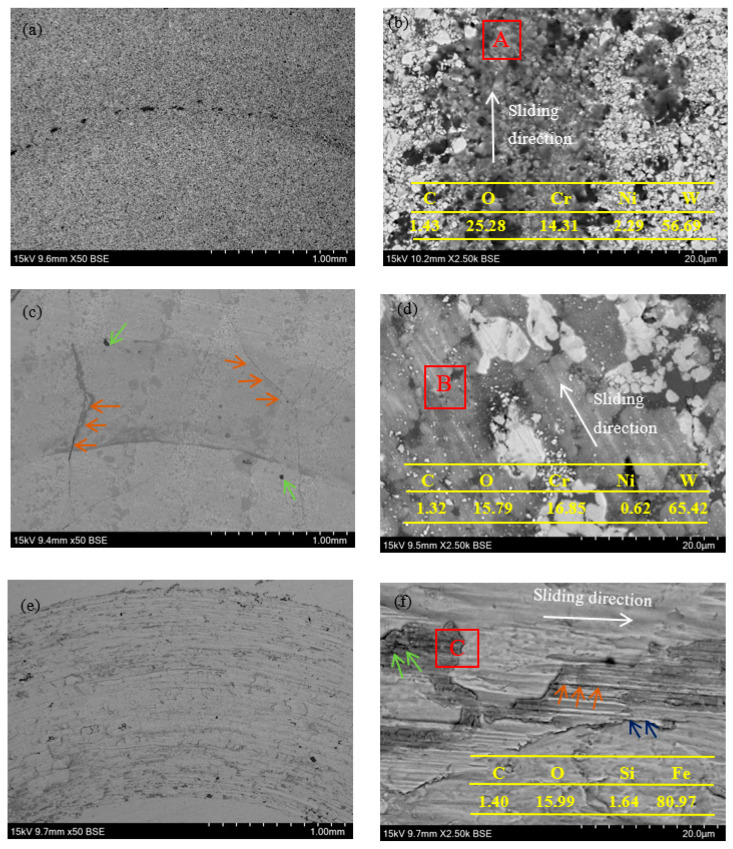
Worn surface micrographs of (**a**,**b**) HVAF-WC coating, (**c**,**d**) LC-WC coating and (**e**,**f**) SK5 steel sample.

**Figure 10 materials-16-02269-f010:**
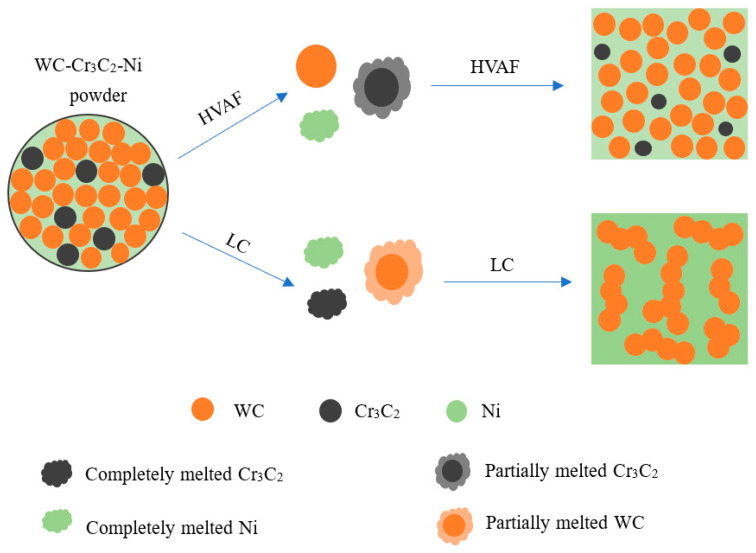
Scheme model of HVAF-WC coating and LC-WC coating formation.

**Figure 11 materials-16-02269-f011:**
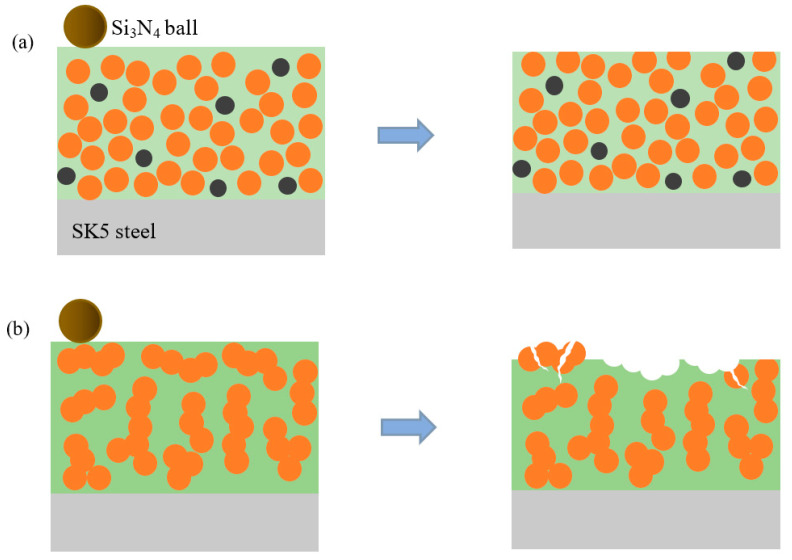
Schematic diagrams of wear mechanism for HVAF-WC coating (**a**) and LC-WC coating (**b**).

**Table 1 materials-16-02269-t001:** Chemical composition of WC–Cr3C2–Ni powder and SK5 steel (wt%).

	WC	Cr_3_C_2_	Ni	C	Si	Mn	S	P	Cr	Cu	Fe
powder	73	20	7	/	/	/	/	/	/	/	/
SK5 steel	/	/	≤0.2	0.8–0.9	0.1–0.35	0.1–0.5	≤0.03	≤0.03	≤0.25	≤0.3	bal

**Table 2 materials-16-02269-t002:** The parameters of HVAF spraying.

Spray Parameter	Unit	Value
Air flow	psi	105
Fuel 1 (propane) flow	psi	101
Fuel 2 (propane) flow	psi	111
Nitrogen carrier gas flow	L/min	60
Spray distance	mm	300

**Table 3 materials-16-02269-t003:** The parameters of Laser Cladding.

Spray Parameter	Unit	Value
Laser power	W	600
Scanning speed	mm/min	600
Powder feeding rate	g/min	14
Overlap	%	50
Nitrogen carrier gas flow	L/min	10
Argon protective gas flow	L/min	10

**Table 4 materials-16-02269-t004:** Chemical composition of the marked sites in Figure 3 by EDS analysis [wt%].

Position	C	Cr	Ni	W
1	2.04 ± 0.37	1.87 ± 0.44	0.33 ± 0.21	95.76 ± 6.01
2	2.72 ± 0.30	53.10 ± 5.55	8.41 ± 0.65	35.77 ± 3.52

**Table 5 materials-16-02269-t005:** Chemical composition of the marked positions in Figure 4 by EDS analysis [wt%].

Position	C	Cr	Fe	Ni	W
1	2.65 ± 0.42	15.39 ± 0.50	/	0.12 ± 0.09	81.84 ± 6.52
2	3.26 ± 0.40	25.42 ± 0.67	52.32 ± 1.20	0.58 ± 0.36	18.42 ± 3.61

## Data Availability

The data presented in this study are available on request from the corresponding author. The data are not publicly available due to privacy.

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
