# Peer review of "Microstructure and Tribo-Behavior of WC–Cr3C2–Ni Coatings by Laser Cladding and HVAF Sprayed: A Comparative Assessment"

_materials, 2023, doi:10.3390/ma16062269_

Round 1

Reviewer 1 Report

1) What is the main significance of paper in comparison to other published works?

Include statements on the significance of this research at the end of the introduction section.

 2) Please correct "about 57-62HRC" as "about 57-62 HRC".

 3) Please correct Figure 1. The insert on Figure 1 is poor quality, which makes it difficult to read and understand the manuscript.

 4) The figures in whole manuscript are of poor quality, which leads to difficulty in reading and understanding the manuscript. Authors must provide high-quality images and illustrations for better clarification and readership. Please correct the format of symbols a), b), c), etc. and symbols, for example, sliding direction, A, B, etc.

5) Please correct "Table 4. Chemical composition of the marked sites in Fig. 2 by EDS analysis [wt%]" as " Table 4. Chemical composition of the marked sites in Fig. 3 by EDS analysis [wt%]"

5) Results and discussion: - To increase the scientific value of the manuscript Authors should consider extension of the all results section with comparison of obtained results with the results described in previous publications.

Reviewer 2 Report

The article entitled “Microstructure and Tribo-behavior of WC–Cr3C2–Ni coatings 2 by laser cladding and HVAF sprayed: A comparative assessment" presents the comparison properties of WC–Cr3C2–Ni coating deposited on the SK5 steel substrate by using High-velocity air fuel spray (HVAF) and 12 Laser cladding (LC). The subject is interesting for readers of Materials journal. However, in order to accept the publication some improvements are recommended:

·           2. Experimental

The methodology of surface treatment is clearly. I would only add information about the measurement error of the chemical composition of the powders used for the coating production process (Table 1).

·           3. Results

Phase Composition and Microstructure.

Figure 1 is unreadable. In my opinion, the EDS analysis diagram is illegible. It should be presented in a separate figure.

In the description of Figure 3 and 4, the numerical value of magnification should be given.

In table 4 and 5 containing the chemical composition of the coatings, the measurement error should be given.

Wear and Friction bahavior

The figure 7 is unreadable. Maybe it's a good idea to present the results of hardness tests and the coefficient of friction in separate drawings.

Please check if the axis descriptions in figure 7 c, d, e are correct. In my opinion, the wear track should be rearranged in the same scale and additionally for coatings in magnification..

·         Stylistic remarks

The article should be stylistically corrected. Sample fixes are shown below:

-     The descriptions of the figures 7need to be changed. In the description of the figures, first should be written the information on what the figure refers, and then information on what the symbols a, b, c mean.

-     References should be standardized in accordance with the requirements of the journal.
